# Synthesis, Characterization, and Antioxidant Evaluation of Novel Pyridylurea-Functionalized Chitosan Derivatives

**DOI:** 10.3390/polym11060951

**Published:** 2019-06-01

**Authors:** Jingjing Zhang, Wenqiang Tan, Lijie Wei, Fang Dong, Qing Li, Zhanyong Guo

**Affiliations:** 1Key Laboratory of Coastal Biology and Bioresource Utilization, Yantai Institute of Coastal Zone Research, Chinese Academy of Sciences, Yantai 264003, Shangdong, China; jingjingzhang@yic.ac.cn (J.Z.); wqtan@yic.ac.cn (W.T.); ljwei@yic.ac.cn (L.W.); fdong@yic.ac.cn (F.D.); 2Center for Ocean Mega-Science, Chinese Academy of Sciences, 7 Nanhai Road, Qingdao 266071, China; 3University of Chinese Academy of Sciences, Beijing 100049, China

**Keywords:** antioxidant ability, chitosan derivatives, pyridylurea

## Abstract

In order to improve the bioactivity of chitosan, we synthesized a novel series of chitosan derivatives: firstly, chitosan was reacted with methylclhlorofonmate obtaining *N*-methoxyformylated chitosan (1), which was then converted into *N*-pyridylurea chitosan derivatives (2a-2c) by amine-ester exchange reaction. In addition, *N*-pyridylurea chitosan derivatives were conducted by reacting with iodomethane to obtain quaternized *N*-pyridylurea chitosan derivatives (3a-3c). The structural characteristics of as-prepared chitosan derivatives were confirmed by fourier transform infrared (FT-IR), ^1^H nuclear magnetic resonance (^1^H NMR), elemental analysis, and scanning electron microscope (SEM). Meanwhile, the antioxidant activity of the chitosan derivatives was assessed in vitro. As shown in this paper, the antioxidant activity decreased in the order: c > b > a. Moreover, after the quaternization with iodomethane, quaternized *N*-pyridylurea chitosan derivatives immediately exhibited enhanced antioxidant capacity compared with *N*-pyridylurea chitosan derivatives. For example, in 1,1-Diphenyl-2-picrylhydrazyl (DPPH) radical scavenging assay, the scavenging activities of 3a-3c were 91.75%, 93.63%, and 97.63% while 2a-2c were 42.32%, 42.97%, and 43.07% at 0.4 mg/mL. L929 cells were also adopted for cytotoxicity test of chitosan and synthesized derivatives by CCK-8 assay and all samples showed decreased cytotoxicity. These results suggested that the novel pyridylurea-functionalized chitosan derivatives could be an ideal biomaterial.

## 1. Introduction

Free radicals generated from oxidation reactions may occur in various metabolic processes which can damage cell membranes, lipids, cellular proteins, and DNA, leading to heart, kidney, liver-related diseases such as liver damage, myocardial infarction, cancer, stroke, Parkinson’s disease, and Alzheimer’s disease [1,2,3]. Antioxidant, a class of molecule inhibiting the oxidation of other molecules, can terminate chain reactions thereby avoiding oxidation and thus beneficial for health effects [4,5]. Therefore, discovering new synthetic antioxidants is of great significance in the fields of antioxidant synthesis and organic transformations.

Chitosan, Poly (β-(1→4)-2-amino-2-deoxy-d-glucan, is generally produced via deacetylation of chitin [6,7,8]. Due to its unique cationic properties, it has attracted wide attention in various fields [9]. For example, chitosan has been employed in applications ranging from food and nutrition to cosmetics, wound healing, wastewater treatment, artificial skin, ophthalmology, and antimicrobial agents [10,11,12,13]. Over the past decade, the number of publications in these areas has increased more than 10 times because of the excellent properties of chitosan, such as biodegradability and biocompatibility as well as the desirable characteristics provided by a variety of chemical modifications [14,15,16]. Exactly, chemical modification is a desirable approach to expand the utilization of chitosan as its bioactivity cannot meet the needs of commercial development [17,18,19]. Therefore, a series of chitosan derivatives with excellent antioxidant activity synthesized by chemical modification was reported in the current research. Some of them were expected to be developed as antioxidants for human consumption.

Pyridine, exhibiting excellent bioactivities, such as antibacterial, antioxidant, antitumor, analgesic, and antiparasitic activities, offers a wide range of applications in agrochemical, pharmaceutical, and fine chemical industries [20,21,22]. In particular, pyridine derivatives could improve the physicochemical activities, solubility, and biological properties of polysaccharide and could be applied in antioxidant and biomedical applications [23,24]. Meanwhile, the improved antioxidant activity of urea derivatives had been reported in the previous publications [25,26,27]. Consequently, pyridylurea would be a satisfactory group that possesses great biological properties. However, there are just a few researches on antioxidant activity of chitosan derivatives bearing pyridylurea as well as quaternized N-pyridylurea chitosan derivatives.

Bai et al. had reported that the amino group in chitosan could be almost completely converted to ureido group with the condition of a large excess of methyl chloroformate and primary amine of low stereohindrance [28,29]. Therefore, a series of *N*-pyridylurea chitosan derivatives (2a-2c) were synthesized on the basis of this design idea. Furthermore, we synthesized quaternized *N*-pyridylurea chitosan derivatives (3a-3c) since the exposed pyridine was vulnerable to alkylating agents to obtain *N*-pyridinium salts, which is conducive to improving the antioxidant activity of chitosan compounds through retarding the formation of free radicals or stabilizing free radicals [24,30]. The chemical structures of the chitosan derivatives were characterized by FT-IR, ^1^H NMR, and elemental analysis. Meanwhile, the morphology of samples was analyzed by scanning electron microscope (SEM). Furthermore, three common free radicals, including DPPH radical, superoxide radical, and hydroxyl radical, and reducing power were selected to evaluate the antioxidant property of chitosan and chitosan derivatives synthesized in vitro. The cytotoxic effects of all samples were also evaluated on L929 cells by CCK-8 assay. These targeted chitosan derivatives have the advantage of high antioxidant activity as well as low cytotoxicity and can be used as potential antioxidants. Moreover, the relationship between the structure and antioxidant activity of chitosan derivatives was analyzed briefly.

## 2. Materials and Methods

### 2.1. Materials

Chitosan with a molecular weight of 5–8 kDa and degree of deacetylation of 73.5% was supplied by Golden-Shell Pharmaceutical Co. Ltd. (Zhejiang, China). Iodomethane, 2-aminopyridine, 3-aminopyridine, and 4-aminopyridine were purchased from the Sigma-Aldrich Chemical Corp. Methyl chloroformate, methanol, triethylamine, *N*,*N*-dimethyl acetamide (DMAc), *N*-Methyl pyrrolidone (NMP), and lithium chloride were supplied by Sinopharm Chemical Reagent Co., Ltd., Shanghai, China.

### 2.2. Analytical Methods

#### 2.2.1. Fourier Transform Infrared (FT-IR) Spectroscopy

Jasco-4100 FT-IR spectrometer (Japan, provided by JASCO Co., Ltd., Shanghai, China) was used to confirm the structure of chitosan derivatives preliminarily. Its scan range is 4000–400 cm^−1^ with a resolution of 4.0 cm^−1^. All samples were grinded and mixed separately with KBr in a ratio of 1:100 for testing.

#### 2.2.2. Nuclear Magnetic Resonance (NMR) Spectroscopy

^1^H nuclear magnetic resonance spectrometer was measured using a Bruker AVIII-500 Spectrometer (500 MHz, Switzerland, provided by Bruker Tech. and Serv. Co., Ltd., Beijing, China) at 25 °C using Deuterium Oxide (D_2_O) as solvents. It was an effective method to determine the functionalization of chitosan derivatives.

#### 2.2.3. Elemental Analysis

The elemental analysis was performed on an element analysis instrument (Vario Micro Elemental Analyzer, Elementar, Germany). The carbon-nitrogen ratios were used to evaluate the degree of substitution in chitosan derivatives and the degrees of substitution (DS) were calculated by the following equations:$$DS_{1}=\frac{n_{1}\times M_{C}-M_{N}\times W_{C/N}}{n_{2}\times M_{C}}$$
$$DS_{2}=\frac{M_{N}\times W_{C/N}+n_{2}\times M_{C}\times DS_{1}-n_{1}\times M_{C}}{n_{3}\times M_{C}}$$
$$DS_{3}=\frac{n_{1}\times M_{C}+n_{3}\times M_{C}\times DS_{2}-n_{1}^{\prime }\times M_{N}\times W_{C/N}-n_{2}\times M_{C}\times DS_{1}}{n_{2}^{\prime }\times M_{N}\times W_{C/N}-n_{4}\times M_{C}}$$
$$DS_{4}=\frac{n_{1}^{\prime }\times M_{N}\times W_{C/N}+n_{2}^{\prime }\times M_{N}\times W_{C/N}\times DS_{3}+n_{2}\times M_{C}\times DS_{1}-n_{1}\times M_{C}-n_{3}\times M_{C}\times DS_{2}-n_{4}\times M_{C}\times DS_{3}}{n_{5}\times M_{C}}$$
where *DS*_1_, *DS*_2_, *DS*_3_, and *DS*_4_ represent the deacetylation degree of chitosan, the DS of derivative 1, the DS of derivatives 2a-2c, and the DS of derivatives 3a-3c; *M_C_* and *M_N_* are the molar mass of carbon and nitrogen, *M_C_* = 12, *M_N_* = 14; *n*_1_, *n*_2_, *n*_3_, *n*_4_, and *n*_5_ are the number of carbon of chitin, chitosan, derivative 1, derivatives 2a-2c, and derivatives 3a-3c, *n*_1_ = 8, *n*_2_ = 2, *n*_3_ = 2, *n*_4_ = 4, *n*_5_ = 1; *n*^′^_1_ and *n*^′^_2_ are the number of nitrogen of chitosan and derivatives 2a-2c, *n*^′^_1_ = 1, *n*^′^_2_ = 2; *W_C/N_* represents the mass ratio between carbon and nitrogen in chitosan derivatives.

#### 2.2.4. Scanning Electron Microscope (SEM)

Scanning electron microscope (SEM) (S-4800, Hitachi, Japan) was used to check the morphology of prepared samples. Before being scanned and photographed, each sample was coated with gold in an ion sputter (E-1045, Hitachi, Japan). During image acquisition, the accelerating potential was 3 kV and the magnification was 1000 times.

### 2.3. Synthesis of Chitosan Derivatives

#### 2.3.1. Synthesis of *N*-Methoxyformylated Chitosan (1)

Chitosan (2.0 g, 12.4 mmol) was dissolved in 30 mL of deionized water, and the solution was cooled in an ice bath. Then, methanol (70 mL) was added when the temperature of the solution was below 10 °C. When the temperature was below 5 °C, methyl chloroformate (7.68 mL, 99.2 mmol) was added with stirring. The mixture was stirred for 7 h at 2–10 °C. During this period, the reaction mixture was maintained in a range of pH 2–7 by adding triethylamine. Upon reaction completion, the product was filtered and washed entirely with excess ethanol. After a freeze-drying overnight in vacuum, *N*-methoxyformylated chitosan (1) was obtained.

#### 2.3.2. Synthesis of *N*-Pyridylurea Chitosan Derivatives (2a-2c)

The prepared *N*-methoxyformylated chitosan (0.44 g, 2 mmol) was dissolved in a solution of LiCl in DMAc (with a mass concentration of 8%). Then, aminopyridine (1.51 g, 16 mmol) was added and the solution was stirred for 12 h at 110 °C. After the reaction, the solution was poured into ethanol to afford some precipitate and the precipitate was filtered and washed with ethanol. Finally, *N*-pyridylurea chitosan derivatives were achieved after vacuum freeze-drying for 24 h.

#### 2.3.3. Synthesis of Quaternized *N*-Pyridylurea Chitosan Derivatives (3a-3c)

0.56 g above synthesized *N*-pyridylurea chitosan derivatives were dispersed into 30 mL of NMP for 12 h at room temperature (r.t.). Then, the reaction was carried out at 60 °C for 24 h with reflux stirring after 0.4 mL CH_3_I were added. After the completion of the reaction, the solution was precipitated by excess acetone. At last, the filtered precipitations were purified in a Soxhlet apparatus with ethanol for two days and the unreacted iodomethane and other outgrowth were extracted in this way.

### 2.4. Antioxidant Assays

#### 2.4.1. DPPH-Radical Scavenging Activity Assay

The procedure of DPPH scavenging property testing of the products was as follows [2]: 2 mL of DPPH ethanol solution (180 μmol/L) and different concentrations of testing samples were incubated for 20 min at room temperature. Then, the absorbance of the remained DPPH radical against a blank was estimated at 517 nm. Three replicates were performed on each sample and the DPPH radical scavenging effect was calculated according to the following equation:$$\mathrm{Scavenging}\;\mathrm{effect}\;\left(\%\right)=\left[1-\frac{A_{\mathrm{sample}\;517\;\mathrm{nm}}-A_{\mathrm{control}\;517\;\mathrm{nm}}}{A_{\mathrm{blank}\;517\;\mathrm{nm}}}\right]\times 100$$
where *A*_sample 517 nm_ is the absorbance of the samples, *A*_control 517 nm_ is the absorbance of the control (ethanol replaced DPPH), and *A*_blank 517 nm_ is the absorbance of the blank (distilled water replaced samples). Vitamin C was used as a positive control.

#### 2.4.2. Superoxide-Radical Scavenging Activity Assay

The procedure of superoxide-radical scavenging ability testing was assessed following the model of Tan’s methods [4]. The reaction mixture (3 mL), involving chitosan or chitosan derivatives (1.5 mL, the concentration was 0.2, 0.4, 0.8, 1.6, and 3.2 mg/mL, respectively), nicotinamide adenine dinucleotide reduced (NADH, 0.5 mL), nitro blue tetrazolium (NBT, 0.5 mL), and phenazine methosulfate (PMS, 0.5 mL) in Tris–HCl buffer (pH 8.0), was incubated at 25 °C for 5 min. The absorbance against a blank was read at 560 nm. Three replicates were performed on each sample and the superoxide-radical scavenging effect was calculated according to the following equation:$$\mathrm{Scavenging}\;\mathrm{effect}\;\left(\%\right)=\left[1-\frac{A_{\mathrm{sample}\;560\;\mathrm{nm}}-A_{\mathrm{control}\;560\;\mathrm{nm}}}{A_{\mathrm{blank}\;560\;\mathrm{nm}}}\right]\times 100$$
where *A*_sample 560 nm_ is the absorbance of the samples, *A*_control 560 nm_ is the absorbance of the control (distilled water replaced NADH), and *A*_blank 560 nm_ is the absorbance of the blank (distilled water replaced samples). Vitamin C was used as a positive control.

#### 2.4.3. Hydroxyl-Radical Scavenging Activity Assay

The test of hydroxyl-radical scavenging ability was carried out according to Hu’s methods with minor modification [20]. The reaction mixture (4.5 mL), containing testing samples of chitosan or chitosan derivatives (1 mL, 0.45, 0.9, 1.8, 3.6, and 7.2 mg/mL), was incubated with EDTA-Fe^2+^ (0.5 mL, 220 μM), safranine O (1 mL, 0.23 μM), and H_2_O_2_ (1 mL, 60 μM) in potassium phosphate buffer (pH 7.4) for 30 min at 37 °C. The absorbance of the mixture was measured at 520 nm. Three replicates were performed on each sample and the hydroxyl-radical scavenging effect was calculated according to the following equation:$$\mathrm{Scavenging}\;\mathrm{effect}\;\left(\%\right)=\frac{A_{\mathrm{sample}\;520\;\mathrm{nm}}-A_{\mathrm{blank}\;520\;\mathrm{nm}}}{A_{\mathrm{control}\;520\;\mathrm{nm}}-A_{\mathrm{blank}\;520\;\mathrm{nm}}}\times 100$$
where *A*_sample 520 nm_ is the absorbance of the samples, *A*_control 520 nm_ is the absorbance of the control (potassium phosphate buffer replaced H_2_O_2_), and *A*_blank 520 nm_ is the absorbance of the blank (distilled water replaced samples). Vitamin C was used as a positive control.

#### 2.4.4. Reducing Power Assay

The procedure of reducing power testing was determined according to the method of Xing with minor modification [31]. 1 mL of different concentrations of testing samples (0.6, 1.2, 2.4, 4.8, and 9.6 mg/mL) in phosphate buffer (200 μmol/L, pH 6.6) were mixed with 1 mL of 1% potassium ferricyanide. The mixture was incubated at 50 °C for 20 min. Then, the reaction was terminated by trichloroacetic acid (10%, *w*/*v*). After centrifuged at 3000 rpm for 10 min, 1.5 mL of the upper layer of solution was mixed with 1.2 mL of distilled water and 0.3 mL of ferric chloride (0.1%, *w*/*v*), and the absorbance was recorded at 700 nm. Higher absorbance of the mixture indicated higher reducing power.

### 2.5. Cytotoxicity Assay

The cytotoxicity of chitosan and synthesized chitosan derivatives a–e on mouse fibroblasts (L929) cells at different concentrations (1.0, 10.0, 100.0, 500.0, and 1000.0 μg/mL) was determined by Cell Counting Kit-8 (CCK-8) assay in vitro. Briefly, the reagent contained wst-8, which was reduced to water-soluble Formazan dye by dehydrogenase in cells under the action of 1-Methoxy PMS. The amount of Formazan dye is proportional to the number of living cells. Therefore, this property can be directly used for cell proliferation and cytotoxicity analysis. The experimental operation is as follows: L929 cells were cultured in RPMI medium (containing 1% mixture of penicillin & streptomycin and 10% fetal calf serum) at 37 °C. These cells were seeded on 96-well flat-bottom culture plates at a density of 1.0 × 105 cells and incubated (37 °C, 5% CO_2_). After 24 h of cell attachment, the samples with different final concentrations were introduced to cells, separately. Next, the cells were cultured for 24 h. Afterward, 10 μL of CCK-8 solution was added in each well and incubated for another 4 h at 37 °C. The absorbance at 450 nm was recorded using a microplate reader. Cell viability was recorded using the following formula:$$\mathrm{Cell}\;\mathrm{viability}\;\left(\%\right)=\frac{A_{\mathrm{sample}\;}-A_{\mathrm{blank}}}{A_{\mathrm{negative}\;}-A_{\mathrm{blank}}}\times 100$$
where *A*_sample_ is the absorbance of the samples (containing cells, CCK-8 solution, and sample solution), Ablank is the absorbance of the blank (containing RPMI medium and CCK-8 solution), and Anegative is the absorbance of the negative (containing cells and CCK-8 solution).

### 2.6. Statistical Analysis

All data were expressed as means ± the standard deviation (SD, *n* = 3) from three replicates. Significant differences were conducted using Scheffe’s multiple range test. *P* < 0.05 was considered statistically significant.

## 3. Results and Discussion

### 3.1. Chemical Synthesis and Characterization

The synthetic strategy of the novel pyridylurea-functionalized chitosan derivatives is shown in Scheme 1. Firstly, *N*-methoxyformylated chitosan (1) was carried out in a solution containing methanol and methyl chloroformate. At the same time, in order to avoid the hydrolysis of methyl chloroformate and *N*-methoxyformylated chitosan, the reaction must be performed under conditions of lower temperature and weak acidity. Then, *N*-pyridylurea chitosan derivatives (2a-2c) were synthesized through the reaction of *N*-methoxyformylated chitosan and aminopyridine at 110 °C. Furthermore, the preparation of quaternized *N*-pyridylurea chitosan derivatives (3a-3c) was accomplished by quaternization of *N*-pyridylurea chitosan derivatives with iodomethane. FT-IR, ^1^H NMR, elemental analysis, and SEM were performed after each step of the synthesis. The FT-IR, ^1^H NMR, and SEM of chitosan and chitosan derivatives are shown in Figure 1, Figure 2 and Figure 3, respectively. Meanwhile, the yields and the degrees of substitution of chitosan derivatives are shown in Table 1.

#### 3.1.1. FT-IR Spectra

Chitosan and synthesized chitosan derivatives were analyzed using FT-IR and the results were depicted in Figure 1. The spectrum of chitosan shows characteristic absorption bands at approximately 3417 cm^−1^ (O–H and N–H stretching of hydroxyl and amine groups), 2919 cm^−1^, 2881 cm^−1^ (C–H stretching of hydrocarbon), 1596 cm^−1^ (–NH_2_ stretching of amino group), and 1072 cm^−1^ (C–O stretching vibration) [32,33,34]. As to spectrum of *N*-methoxyformylated chitosan (1), the absorbance of amino group at 1596 cm^−1^ disappears and new absorbtion band appears at 1700 cm^−1^ (–NHCOOMe), revealing that the amino group of chitosan was replaced by methoxyformylamino group [28]. For *N*-pyridylurea chitosan derivatives (2a-2c), the absorbtion band of methoxyformylamino group (1700 cm^−1^) disappears when pyridylurea group formed and new absorption bands at about 1640 cm^−1^, 1540 cm^−1^, and 1390 cm^−1^ appear. The bands at 1540 cm^−1^ and 790 cm^−1^ are attributed to the typical absorption of pyridine [20,35] and the band found at 1640 cm^−1^ is attributed to the vibration of the ureido (–NH–CO–NH–) [28]. After *N*-alkylation with iodomethane, the absorption of N-CH_3_ is at 1454 cm^−1^ for quaternized *N*-pyridylurea chitosan derivatives (3a-3c) [36], while the characteristic absorbtion bands of *N*-pyridylurea chitosan derivatives still exist. The above analysis preliminarily proves that *N*-methoxyformylated chitosan, *N*-pyridylurea chitosan, and quaternized *N*-pyridylurea chitosan were synthesized successfully.

#### 3.1.2. NMR Spectra

Figure 2 shows the ^1^H NMR spectra of chitosan and chitosan derivatives. The proton assignment of unmodified chitosan is as follows: the chemical shifts at 3.0 ppm, 3.6–3.9 ppm, and 4.6 ppm are attributed to the hydrogen protons of the chitosan molecule on C-2, C-3 to C-6, and C-1, respectively [19]. For the ^1^H NMR spectrum of *N*-methoxyformylated chitosan, a singlet at 3.69 ppm indicates the presence of a methoxy group [28]. As to 2a-2c, they exhibit characteristic resonances of pyridine ring at about 7.2–8.5 ppm [35]. In addition to all characteristic proton signals of *N*-pyridylurea chitosan derivatives, the ^1^H NMR spectra of 3a-3c show the prominent peak of –N^+^CH_3_ at about 4.23 ppm [24]. In brief, it is enough to confirm that the chitosan derivatives were obtained successfully.

#### 3.1.3. Degrees of Substitution

The degrees of substitution (DS) for *N*-pyridylurea chitosan derivatives and quaternized *N*-pyridylurea chitosan derivatives were estimated by elemental analysis and the results are shown in Table 1. As is shown, the DS of chitosan derivatives increase with the decrease of steric hindrance. Derivatives 2c and 3c with lower steric resistance have higher degree of substitution. For example, the DS of 2c and 3c are 0.244 and 0.590, respectively, whereas the DS of 2a and 3a are only 0.223 and 0.409. The active groups-pyridylurea or quaternized pyridylurea are key factor affecting the antioxidant activity and the bioactivity of these derivatives could be affected directly by the degrees of substitution of these active groups. The higher degree of substitution means more functional groups and higher bioactivity [17,37].

#### 3.1.4. Morphology Analysis

Scanning electron microscope (SEM) is one of the most effective tools to study the surface phenomena of the prepared samples. The images of the surface morphology of pure chitosan (CS), *N*-methoxyformylated chitosan (1), *N*-pyridylurea chitosan derivatives (2a-2c), and quaternized *N*-pyridylurea chitosan derivatives (3a-3c) are shown in Figure 3. The pure chitosan granules are similar in size (approximately 50 to 100 nm) and they have a smooth but not compact surface. It is noticed that after the reaction with methyl chloroformate, *N*-methoxyformylated chitosan shows an irregular and rough surface. The morphology of *N*-pyridylurea chitosan derivatives (2a-2c) is loose particles with varying sizes (approximately 50 to 500 nm). Moreover, while derivatives 2a-2c display nonporous and rough phase surface, derivatives 3a-3c exhibit highly porous. After the quaternization, the formation of cracks appeared in quaternized *N*-pyridylurea chitosan derivatives. These observations show that the microstructures of chitosan changed after chemical modification.

### 3.2. Antioxidant Activity

#### 3.2.1. Scavenging Ability of DPPH Radical

Figure 4 shows the DPPH radical scavenging activity of chitosan and chitosan derivatives. In this test, vitamin C (Vc) was used as a positive control. Obviously, the scavenging properties of chitosan derivatives increase with the rise of concentration. Chitosan and 1 show relatively weak scavenging activity against DPPH radical, with the scavenging indices of 7.92% and 39.13% at 1.6 mg/mL. However, chitosan derivatives 2a-2c and 3a-3c possess much stronger DPPH radical scavenging ability and the scavenging ability of 3a-3c is more efficient than 2a-2c. For example, while the scavenging rates of 2a-2c are 77.45%, 75%, and 76.32%, 3a-3c are 94.12%, 96.01%, and 97.81% at 0.8 mg/mL. Results above demonstrate that the *N*-pyridine position and quaternarization could have different influence on the antioxidant property of chitosan derivatives and the mechanism will be discussed briefly later.

#### 3.2.2. Scavenging Ability of Superoxide Radical

The results of superoxide radical scavenging rates of chitosan and chitosan derivatives at various concentrations are given in Figure 5. The scavenging rates of chitosan derivatives presents several obvious rules, which are c > b > a and 3 > 2. For instance, at the concentration of 1.6 mg/mL, the superoxide radical scavenging rates of 3a-3c (87.88%, 95.52%, and 96.05%) are close to 100%. And the scavenging property of 3a-3c is higher than 2a-2c (50.33%, 60.48%, and 62.52%), 1 (49.75%), and CS (34.72%). The results suggest that quaternized *N*-pyridylurea chitosan derivatives can be investigated as a kind of effective superoxide radical scavenger.

#### 3.2.3. Scavenging Ability of Hydroxyl Radical

The hydroxyl radical scavenging ability of chitosan and synthesized derivatives is shown in Figure 6. The results demonstrate that chitosan and vitamin C have very weak scavenging ability against hydroxyl radical at any tested concentration. However, 2a-2c and 3a-3c could efficiently scavenge more than 80% of hydroxyl radicals at 0.8 mg/mL. Furthermore, all samples show a dose-dependent hydroxyl radical elimination effect.

#### 3.2.4. Reducing Power

The antioxidant activity of chitosan and chitosan derivatives was also evaluated using reducing power assay and the result is presented in Figure 7. Figure 7 shows that vitamin C is the best reducing agent and its absorbing ability is up to 4 A. Of all the synthesized products, 3c and 2c exhibit better activity, which is close to 4 A at 1.6 mg/mL. Meanwhile, the reduced ability of the samples still follows the rules of c > b > a and quaternized *N*-pyridylurea chitosan derivatives > *N*-pyridylurea chitosan derivatives.

Based on the analyses mentioned above, some conclusions could be gained from the antioxidant effects of the products as follows: firstly, for all tested samples, the antioxidant rates are increased along with the enhancement of concentration. As can be seen from Figure 4, Figure 5, Figure 6 and Figure 7, this trend is obvious and applicable to all samples. Secondly, all of the synthesized products have better ability of scavenging DPPH radical, superoxide radical, hydroxyl radical, and reducing power compared with chitosan. It is attributed to the formation of the active groups-pyridylurea or quaternized pyridylurea, which can exert the effects on oxidation resistance through damage the free radical chain reaction. Thirdly, regardless of *N*-pyridylurea chitosan derivatives or quaternized *N*-pyridylurea chitosan derivatives, the antioxidant activities of all samples decrease in the order: c > b > a, which is related to the sort order of the *N*-pyridine position (4-position > 3-position > 2-position). The different *N*-pyridine position could influence the steric hindrance of aminopyridine. Meanwhile, steric hindrance is one of the crucial factors affecting the interaction between aminopyridine and *N*-methoxyformylated chitosan. Generally, the larger steric hindrance leads to a lower degree of substitution (DS) of chitosan derivative. For example, because the nitrogen and amino of 2-aminopyridine are adjacent to each other, it causes high steric hindrance and hinders the reaction of amino and methoxycarbonyl groups. Thus, the degrees of substitution of derivatives 2a and 3a are rather lower, which can be confirmed by the data in Table 1. Accordingly, 2a and 3a have fewer effective components at the same concentration, leading to a relative lower antioxidant activity. However, when amino locates at 4-position, the repulsive interaction is rather lower and the DS is higher. And the antioxidant property of 2c and 3c is higher. Hence, the antioxidant rule of chitosan derivatives is as follow: c > b > a. Fourthly, the antioxidant activity of quaternized *N*-pyridylurea chitosan derivatives is better than the corresponding *N*-pyridylurea chitosan derivatives, which is 3a > 2a, 3b > 2b, and 3c > 2c. In general, the density of positive charge has an effect on the antioxidant activity of compounds, because the positive charge can attract single electron of free radicals to inhibit its chain reaction. Consequently, the quaternized *N*-pyridylurea chitosan derivatives with high-density positive charges would attract more single electron of free radicals, which could perform better antioxidant ability than the corresponding *N*-pyridylurea chitosan derivatives [24]. In summary, it is reasonable to presume that the *N*-pyridylurea is a significant factor influencing the antioxidant activity of samples and the structure-activity relationship would be further researched.

### 3.3. Cytotoxicity Analysis

Biocompatibility is a complex characteristic combining a range of individual biological properties which are preferably tested using alternative in vitro methods. Cytotoxicity assay is a kind of in vitro assay which simulates the living environment of cells in vitro and detects the biological property of cells after contacting samples. Acute cytotoxicity in biomaterials can be evaluated by this assay. It is one of the important test indexes in the biocompatibility evaluation system of biomaterials. Mouse fibroblasts (L929) cells are a well-known and sensitive model to evaluate in vitro cytotoxicity, and it has been extensively used to evaluate toxicity of the biomaterials [38]. In the present work, we used this culture model to evaluate the biocompatibility of chitosan and its derivatives at different concentrations.

After treatment with chitosan and its derivatives for 24h, the growth of L929 cells is shown in Figure 8. At the tested concentration, the cell viabilities of chitosan are about 100%, which means that pristine chitosan has low cytotoxicity. Similarly, derivative 1 is also not cytotoxic. As for derivatives 2a-2c, derivative 2a exhibits weak cytotoxicity. For example, the cell viability of derivative 2a is 82.19% at the highest test concentration. After quaternization, the cell viabilities of derivatives 3a-3c decreased. It seems that positively charged chitosan derivatives bearing quaternary ammonium salts have a slight inhibitory effect on cell growth. The positively charged chitosan derivatives bearing quaternary phosphonium salts could facilitate affinity with the negatively charged components on the extracellular membrane through electrostatic interaction and it is the major cause for cytotoxicity. Considering the excellent antioxidant activities and slightly cytotoxicity of these chitosan derivatives, they might still be considered to have good biocompatibility. Hence, these novel chitosan derivatives could be an ideal biomaterial in fields of food and cosmetics.

## 4. Conclusions

Nowadays, antioxidant polymers have been widely studied, particularly chitosan antioxidants. However, there are very few reports on the antioxidant activity of chitosan derivatives containing pyridylurea. In this paper, a group of pyridylurea-functionalized chitosan derivatives were synthesized successfully. The antioxidant activity and mechanism of synthesized derivatives were investigated. All of the chitosan derivatives exhibit higher antioxidant activity than chitosan. Meanwhile, the order of antioxidant activity, which is c > b > a, illustrates that the *N*-pyridyl position of chitosan derivatives could influence the antioxidant property. Furthermore, the synthesized chitosan derivatives bearing quaternized *N*-pyridylurea showed more significant biological activity than chitosan and chitosan derivatives with *N*-pyridylurea. This activity rule is of great significance for studying the mechanism of antioxidant activity of chitosan derivatives. Meanwhile, the cytotoxicity of chitosan derivatives against L929 cells was assessed by CCK-8 method and the results indicated that the synthetic pyridylurea-functionalized chitosan derivatives had a low cytotoxic effect, which meant that these derivatives had good biocompatibility. These findings support a continued investigation for the safety and non-toxicity of chitosan derivatives. Therefore, these biocompatible pyridylurea-functionalized chitosan derivatives can partly replace traditional antioxidants in fields of food, cosmetic industry, and medicine. To sum up, this study suggests that these designed derivatives will develop the application of chitosan in antioxidant researches and further comprehensive studies to confirm this hypothesis on relations of structure and antioxidant activity will be carried out.

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
