# Peer review of "Synthesis, Characterization, and Antioxidant Evaluation of Novel Pyridylurea-Functionalized Chitosan Derivatives"

_polymers, 2019, doi:10.3390/polym11060951_

Round 1
Reviewer 1 Report
The article is interesting and original and provides new data concerning synthesis and characterization of novel pyridylurea-functionalized chitosan derivatives
However, minor revisions are required as follows:
1. In Abstract, rows 20 and 21 “The structural characteristics of as-prepared chitosan and chitosan derivatives were confirmed by FT-IR, 1H NMR, elemental analysis, and SEM.” Please provide the full name of the characterization methods FT-IR, 1H NMR, and SEM
2. In section 2.5. Cytotoxicity Assay please provide briefly few other information about CCK-8 assay and L929 cells (mouse adipose tissue?)
3. In section 3.1.3. Degrees of Substitution rows 248-249, you said ” Generally, the bioactivity of these derivatives could be affected directly by the degrees of substitution. The higher degree of substitution means more functional groups and higher bioactivity.” Please explain in more details why a higher degree of substitution means more functional groups and higher bioactivity and provide some additional references eventually.
4. In section 3.1.4. Morphology Analysis, Row 259 in phrase containing “………is loose particles with varying sizes” please provide a numerical range of particle size. Please insert a visible scale for all images in Figure 3. SEM images of chitosan and chitosan derivatives
5. In section 3.3. Cytotoxicity Analysis – the choice of L929 cells (mouse adipose tissue?) can be directly related to any biomedical target application? Why this study demonstrates that “novel chitosan derivatives could be an ideal biomaterial in fields of food and cosmetics.” (row 347)
Author Response
Responds to reviewer 1:
Manuscript ID: polymers-504641
Title: Synthesis, characterization, and biological evaluation of novel pyridylurea-functionalized chitosan derivatives as antioxidant agents
Journal: Polymers
Dear reviewer,
Thank you for your comments concerning our manuscript entitled “Synthesis, characterization, and biological evaluation of novel pyridylurea-functionalized chitosan derivatives as antioxidant agents”. Those comments are all valuable and very helpful for revising and improving our paper. We have studied comments carefully and have made corrections which we hope meet with approval. The main corrections in the manuscript are as following:
1. In Abstract, rows 20 and 21 “The structural characteristics of as-prepared chitosan and chitosan derivatives were confirmed by FT-IR, 1H NMR, elemental analysis, and SEM.” Please provide the full name of the characterization methods FT-IR, 1H NMR, and SEM.
Answer: Thank you for your kind suggestions and according to your recommendation, we have provided the full name of the characterization methods of fourier transform infrared (FT-IR), 1H nuclear magnetic resonance (1H NMR), and scanning electron microscope (SEM).
2. In section 2.5. Cytotoxicity Assay please provide briefly few other information about CCK-8 assay and L929 cells (mouse adipose tissue?)
Answer: Thank you for your kind suggestions and we have provided some details about Cell Counting Kit-8 (CCK-8) assay and mouse fibroblasts (L929) cells in Page 5 Lines 186-192.
3. In section 3.1.3. Degrees of Substitution rows 248-249, you said ” Generally, the bioactivity of these derivatives could be affected directly by the degrees of substitution. The higher degree of substitution means more functional groups and higher bioactivity.” Please explain in more details why a higher degree of substitution means more functional groups and higher bioactivity and provide some additional references eventually.
Answer: Thank you for your kind suggestions and according to your recommendation we have provided more details and references on degree of substitution in Page 9 Lines 259-262. Because the active groups-pyridylurea or quaternized pyridylurea are key factor affecting the antioxidant activity, the bioactivity of these derivatives could be affected directly by the degrees of substitution of these active groups. Therefore, the higher degree of substitution means more functional groups and higher bioactivity
4. In section 3.1.4. Morphology Analysis, Row 259 in phrase containing “………is loose particles with varying sizes” please provide a numerical range of particle size. Please insert a visible scale for all images in Figure 3. SEM images of chitosan and chitosan derivatives.
Answer: Thank you for your kind suggestions. We have provided the numerical range of particle size of derivatives 2a-2c (Lines 272-273) and inserted a visible scale for all images in Figure 3.
5. In section 3.3. Cytotoxicity Analysis – the choice of L929 cells (mouse adipose tissue?) can be directly related to any biomedical target application? Why this study demonstrates that “novel chitosan derivatives could be an ideal biomaterial in fields of food and cosmetics.” (row 347)
Answer: Thank you for your kind suggestions. Biocompatibility is a complex characteristic combining a range of individual biological properties which are preferably tested using alternative in vitro methods. Cytotoxicity assay is a kind of in vitro assay which simulates the living environment of cells in vitro and detects the biological property of cells after contacting samples. Acute cytotoxicity in biomaterials can be evaluated by this assay. It is one of the important test indexes in the biocompatibility evaluation system of biomaterials. Mouse fibroblasts (L929) cells are a well-known and sensitive model to evaluate in vitro cytotoxicity, and it has been extensively used to evaluate toxicity of the biomaterials. Lower cytotoxicity means better biocompatibility. Thus, novel chitosan derivatives could be an ideal biomaterial in fields of food and cosmetics. (Lines 349-357).

Reviewer 2 Report
Dear Editor,
The manuscript submitted by Jingjing Zhang et al. and entitled: “Synthesis, Characterization, and Biological Evaluation of Novel Pyridylurea-functionalized Chitosan Derivatives as Antioxidant Agents” is an interesting study which deals with synthesis of Chitosan conjugate with biological activity. This manuscript can be accepted after major revisions. Please to see my comments below.
Comments:
- Authors must rephrase the title. It is too confused.
- In abstract, authors must add data regarding biological activities investigated.
- In introduction part, authors have made lot of statements on chitosan without backed it up with last good references in this field such as reviews on chitosan modifications. Please to add new references in the revised manuscript.
- In material and method, authors must give the units of molecular weight (line 81). In NMR analysis, authors used D20 to solubilize the chitosan. Is it really fully soluble in water? Same thing when author dissolve chitosan in water for the N-methoxyformylated Chitosan. Was it really soluble in water?
- In result part, the data given were not really explained and discussed. Authors just made a very shallow presentation of their results. Then, authors must discuss results in the revised manuscript.
- In the conclusion part, the aspects of novelty and the biological applications should be more underlined.
General comment:
- In the revised manuscript, the authors need to pay more attention to grammatical construction of sentences and spelling of sentences.
- Authors must add the concentration units in the axis of Figures 4-7 in the revised manuscript.
Author Response
Responds to reviewer 2:
Manuscript ID: polymers-504641
Title: Synthesis, characterization, and biological evaluation of novel pyridylurea-functionalized chitosan derivatives as antioxidant agents
Journal: Polymers
Dear reviewer,
Thank you for your comments concerning our manuscript entitled “Synthesis, characterization, and biological evaluation of novel pyridylurea-functionalized chitosan derivatives as antioxidant agents”. Those comments are all valuable and very helpful for revising and improving our paper. We have studied comments carefully and have made corrections which we hope meet with approval. The main corrections in the manuscript are as following:
1. Authors must rephrase the title. It is too confused.
Answer: Thank you for your kind suggestions and according to your recommendation, we have rephrased the title to “Synthesis, characterization, and antioxidant evaluation of novel pyridylurea-functionalized chitosan derivatives”.
2. In abstract, authors must add data regarding biological activities investigated.
Answer: Thank you for your kind suggestions and according to your recommendation we have added data regarding antioxidant activity in Page 1 Lines 25-27.
3. In introduction part, authors have made lot of statements on chitosan without backed it up with last good references in this field such as reviews on chitosan modifications. Please to add new references in the revised manuscript.
Answer: Thank you for your kind suggestions. We have added three reviews ([14]. El Knidri, H.; Belaabed, R.; Addaou, A.; Laajeb, A.; Lahsini, A., Extraction, chemical modification and characterization of chitin and chitosan. Int. J. Biol. Macromol. 2018, 120, (Pt A), 1181-1189. [15]. Mittal, H.; Ray, S. S.; Kaith, B. S.; Bhatia, J. K.; Sukriti; Sharma, J.; Alhassan, S. M., Recent progress in the structural modification of chitosan for applications in diversified biomedical fields. Eur. Polym. J. 2018, 109, 402-434. [16]. Brasselet, C.; Pierre, G.; Dubessay, P.; Dols-Lafargue, M.; Coulon, J.; Maupeu, J.; Vallet-Courbin, A.; de Baynast, H.; Doco, T.; Michaud, P.; Delattre, C., Modification of Chitosan for the Generation of Functional Derivatives. Appl. Sci. 2019, 9, (7), 1321.) and one research paper ([17]. Zhang, J.; Tan, W.; Wei, L.; Chen, Y.; Mi, Y.; Sun, X.; Li, Q.; Dong, F.; Guo, Z., Synthesis of urea-functionalized chitosan derivatives for potential antifungal and antioxidant applications. Carbohydr. Polym. 2019, 215, 108-118.) in References part.
4. In material and method, authors must give the units of molecular weight (line 81). In NMR analysis, authors used D2O to solubilize the chitosan. Is it really fully soluble in water? Same thing when author dissolve chitosan in water for the N-methoxyformylated chitosan. Was it really soluble in water?
Answer: Thank you for your kind suggestions and according to your recommendation we have given the units of molecular weight of chitosan. In 1H NMR analysis, 20 mg sample was dissolved in 0.6 mL D2O for test. Because the molecular weight of chitosan we used is only 5-8 kDa, under this test condition, the chitosan and N-methoxyformylated chitosan can be completely dissolved.
5. In result part, the data given were not really explained and discussed. Authors just made a very shallow presentation of their results. Then, authors must discuss results in the revised manuscript.
Answer: Thank you for your kind suggestions and according to your recommendation, we have enhanced the discussion about degrees of substitution (Lines 259-262), morphology analysis (Lines 272-273), and cytotoxicity analysis (Lines 366-369). Meanwhile, the antioxidant rules of the chitosan derivatives and the mechanism between structure and antioxidant activity are mainly discussed on Lines 318-347.
6. In the conclusion part, the aspects of novelty and the biological applications should be more underlined.
Answer: Thank you for your kind suggestions and according to your recommendation we have increased the discussion on the novelty and the biological applications of the synthesized derivatives in conclusion part (Lines 381-391).
7. In the revised manuscript, the authors need to pay more attention to grammatical construction of sentences and spelling of sentences.
Answer: Thank you for your kind suggestions and according to your recommendation we have revised the grammatical construction and spelling of sentences (Line 34, Lines 59-61, Lines 307-308, and so on).
8. Authors must add the concentration units in the axis of Figures 4-7 in the revised manuscript.
Answer: Thank you for your kind suggestions and according to your recommendation we have added the concentration units in Figures 4-7.

Round 2
Reviewer 2 Report
Authors revised manuscript according to reviewer's comments.
Then, this paper can be published in journal.